# Dietary Sodium and Potassium Intake in Hungarian Elderly: Results from the Cross-Sectional Biomarker2019 Survey

**DOI:** 10.3390/ijerph18168806

**Published:** 2021-08-20

**Authors:** Eszter Sarkadi-Nagy, Andrea Horváth, Anita Varga, Leonóra Zámbó, Andrea Török, Georgina Guba, Nikolett Szilfai, Andrea Zentai, Márta Bakacs

**Affiliations:** Department of Nutritional Epidemiology, National Institute of Pharmacy and Nutrition, Albert Flórián út 3/A, 1097 Budapest, Hungary; horvath.andrea1@ogyei.gov.hu (A.H.); varga.anita@ogyei.gov.hu (A.V.); zambo.leonora@ogyei.gov.hu (L.Z.); torok.andrea@ogyei.gov.hu (A.T.); guba.georgina@ogyei.gov.hu (G.G.); szilfai.nikolett@ogyei.gov.hu (N.S.); zentai.andrea@ogyei.gov.hu (A.Z.); bakacs.marta@ogyei.gov.hu (M.B.)

**Keywords:** Hungary, salt, sodium, potassium, 24 h urinary excretion, elderly

## Abstract

High sodium intake and inadequate potassium intake are associated with high blood pressure. The elderly are more salt sensitive than other age groups, yet a reliable estimate of the dietary sodium and potassium intake of this age group in Hungary is unavailable. The study aimed to estimate the sodium and potassium intakes in the Hungarian elderly from 24 h urine sodium and potassium excretion. In this cross-sectional study, participants were selected from patients of general practitioners practicing in western Hungary. The participants comprised 99 men and 90 women (mean age 67.1 (SD 5.4] years) who participated in the Biomarker2019 survey and returned a complete 24 h urine collection. We assessed dietary sodium and potassium by determining 24 h urinary sodium and potassium excretions and 3-day dietary records. The mean urinary sodium was 188.8 (73.5) mmoL/day, which is equivalent to 11.0 g of salt/day; and the mean urinary potassium was 65.8 (24.3) mmoL/day, which is equivalent to 3.03 g of potassium/day, after adjusting for non-urinary potassium losses. Only 7% of the subjects met the World Health Organization’s recommended target of less than 5 g of salt/day, and 33% consumed at least the recommended potassium amount of 3.5 g/day, based on the estimates from 24 h urine excretion. For most elderly, sodium intake exceeds, and potassium does not reach, dietary recommendations. The results underline the need to intensify salt reduction efforts in Hungary.

## 1. Introduction

Non-communicable diseases (NCDs) comprise the greatest fraction of deaths worldwide, with the total numbers of deaths from NCD continuously increasing [1]. Hungary has a high prevalence of deaths from NCDs, accounting for approximately 94% of all deaths in 2016, with 48% attributable to cardiovascular diseases (CVDs) [2]. High blood pressure (BP) is an important risk factor for CVDs [3,4]. In the Hungarian adult population, the prevalence of raised BP in people aged 18 years or older is 37%; in the elderly (>60 years), the prevalence of hypertension (as diagnosed by a general practitioner) reaches 63% [2]. High salt (NaCl) intake raises BP and increases the risk of developing hypertension [5]. In 2017, the Global Burden of Disease Study reported that high sodium intake is a leading dietary risk factor, contributing to 3 million deaths and 70 million disability-adjusted life years globally [6]. A reduction in population salt consumption is an effective intervention to reduce the burden of hypertension [7] and is one of the cost-effective policy options (‘best buys’) identified in the World Health Organization (WHO) Action Plan [8]. WHO member states agreed to a voluntary global NCD target of a 30% relative reduction in the mean population intake of salt, aiming to achieve less than 5 g of salt/day (approximately 2 g of sodium/day) by 2025 [9,10].

Hypertension is a widespread problem in the elderly, who are more sensitive to the BP-lowering effects of a decreased salt intake than other age groups [11]. Sodium reduction to a level of 1500 mg/day lowers BP more in older adults than in younger adults [12]. Furthermore, a reduction of salt intake increases the efficacy of the renin-angiotensin system blockers in terms of their BP-lowering effects [13]. The elderly, who experience the effects of hypertension and the adverse effects of antihypertensive medications, can benefit considerably from salt intake reduction [14].

Increased potassium intake reduces BP values in adults and improves cardiovascular health [15,16]. Low dietary potassium can increase the effect of sodium on BP, and the relationship between sodium and BP is strengthened if the urinary sodium/potassium ratio is considered [17]. The Hungarian national recommendation coincides with that of the WHO that adults should consume a minimum of 3.5 g (90 mmoL) of potassium daily [18,19].

In light of the cardiovascular implications of consuming too much sodium and too little potassium, an important public health initiative is to monitor the intake of these nutrients [20]. Determining 24 h urinary sodium excretion is considered to be the most accurate method for measuring salt intake [21]. The International Consortium for Quality Research on Dietary Sodium/Salt recommends multiple 24 h urine samples collected over a series of days from a representative population sample to assess an individual’s 24 h dietary sodium intake [22]; however, a single 24 h urine sample is considered sufficient to estimate a population’s mean intake. Potassium intake can also be estimated from a 24 h urine after adjusting for non-urinary losses [18]. To our knowledge, the Intersalt study conducted in 1985–86 was the only 24 h urinary sodium and potassium excretion measurement reported on a large Hungarian sample [23]. The National Diet and Nutritional Status Survey, the main survey monitoring the sodium and potassium intake of the Hungarian population, uses three-day dietary records to estimate sodium and potassium consumption [24,25]. The limitations of dietary assessment methods have been well documented and include the subjective reporting of dietary intake, portion sizes, difficulty in accurately measuring discretionary salt (at the table and during cooking) and inaccuracies in food composition databases [26].

The present study aims to estimate sodium and potassium consumption in a Hungarian elderly sample. For intake estimation, we used the traditional three-day dietary records and the newly established 24 h urine collection, which is considered the gold standard for salt intake estimation [27]. We aimed to apply the new methodology developed during the present study to future studies on a large, nationally represented sample.

## 2. Materials and Methods

### 2.1. Study Design and Recruitment

This study was a part of the Biomarker2019 survey, which involved collecting anthropometric, demographic, dietary and biological marker data from the elderly to assess risk factors for NCD development, nutrition knowledge, and nutritional status. Data from this cross-sectional survey were used to develop age-specific national dietary guidelines for the elderly. The data were collected from 29 May to 7 December 2019 in the western region of Hungary. Eight research sites (general practitioners’ offices) were involved in the recruitment of subjects from six settlements. Participants were selected from general practitioners’ patients who met the inclusion and exclusion criteria. The study was open to persons who were at least 60 years of age, living in Hungary, non-institutionalized, and had signed the informed consent form. The exclusion criteria were severe stage 4 renal insufficiency diagnosed by a physician, diuretic therapy that started within two weeks of the start of the study, acute infectious disease, cancer treatment at the time of data collection, and any other conditions that would compromise the collection of 24 h urine samples. A total of 202 people were included in the study. Two people withdrew from the sample after providing consent. A total of 200 people (104 men and 96 women) completed the questionnaires and dietary records. Eleven participants reported missing more than one void or provided under-collection (<22 h); thus, the final sample size was 189. Using quota sampling, we determined the following targeted sex and age distributions: 50% men and 50% women, aged 60–64 years (35.0% of subjects), 65–69 years (35% of subjects), or ≥70 years (30% of subjects). The sex and age distributions of the final sample were: 68 people aged 60–64 years (36.0% of subjects), 73 people aged 65–69 years (39% of subjects), and 48 people aged ≥70 years (25% of subjects). The survey was carried out in accordance with the Declaration of Helsinki and good clinical practice [28]. The study was approved by the National Center for Public Health on the basis of the resolution of the Scientific and Research Ethics Committee of the Health Science Council with registration number 19700-5/2019/EÜIG.

### 2.2. Data Collection

The data collection was comprised of the following items: (a) questionnaire, (b) physical measurement, (c) three-day dietary record, (d) biological samples (fasting blood and 24 h urine). Twelve registered nurses with health science backgrounds received a one-day training as field staff for data collection. The study involved two separate visits, appointments of which were scheduled at the general practitioners’ offices with the field staff. At the first visit, patient information was provided, consent was signed, a data sheet was prepared to certify compliance with the inclusion and exclusion criteria, tools and guidelines for urine collection were handed over to the participant, and processes of urine collection and the filling of the three-day dietary record were explained. At the second visit, the biological samples were collected (blood was drawn, and the collected 24 h urine was submitted), three-day dietary records were returned, and motivational gifts (voucher) were handed over to the participants. The other elements of the study (anthropometric measurements, conducting the questionnaire) were performed in consultation with the participants, generally during the first visit.

The questionnaire (an adapted version of the European Health Interview Survey/National Diet and Nutrition Status Survey) [29,30] obtained background variables on demographics, socio-economic status, health status, health determinants such as fruit and vegetable consumption, smoking, and alcohol consumption; and a section on knowledge, attitudes and behavior on dietary salt consumption. Anthropometry, BP, and heart rate were measured in all participants with standardized protocols and validated equipment, which are described elsewhere [31].

### 2.3. 24 h Urine Collection

Participants received both verbal and written instructions for 24 h urine collection. Urine was collected during one of the days for diet recording. The WHO protocol was followed for the 24 h urine collection [31]. In brief, subjects were requested to void, discard the first-pass urine, and note the time in the morning of the start of the 24 h period. All urine passed thereafter was collected in the container provided, including the first urine of the following morning, and the final time was recorded. Any missing void was noted. The last urine was requested to be collected at approximately the same time as when the collection started the day before. The collected urine was stored in a provided 3-L screw-top container with the lid on, in a cool, dry area. Urine samples were submitted to the field staff at the second visit and transferred to a local laboratory for volume measurement and analysis. Urine samples were considered incomplete if any of the following occurred: (a) the total 24 h urinary volume was <500 mL; (b) the first void was not discarded but collected at the start of the 24 h period; (c) the time of the first void was not recorded (start of the urine collection); (d) urine was under-collected (<22 h) or over-collected (>26 h); (e) more than one void was not collected. On the basis of these criteria, we excluded eleven samples from the analysis. Sodium and potassium content in the urine were determined through the ion-selective electrode (indirect) method with a Roche Cobas c311 system (catalogue number: ISE indirect Na-K-Cl for Gen.2 11360981 216). Creatinine content was determined through the urinary creatinine enzymatic UV method with the Roche Cobas c311 System (catalogue number: CREP203263991 190). Measurements were carried out at the Medical Laboratory DRC Ltd. Balatonfüred. Values for sodium or potassium (mmoL/24 h) were converted into g/day (1 mmoL = 0.023 g of sodium, 1 mmoL = 0.039 g of potassium) to estimate dietary intake. The sodium value was multiplied by 2.5421 to convert dietary sodium intake into salt (NaCl) intake. We assumed that 100% of sodium and 85% of potassium ingested are excreted in the urine [18]. We elected not to adjust these final sodium estimates for non-urinary losses (e.g., sweat) because urinary excretion accounts for ≥90% of dietary intake on average [32,33].

### 2.4. Dietary Data

Participants received verbal and written instructions for filling out the three-day dietary records. Instructions included details on how to estimate portion size, how to specify foods and drinks consumed (including the brand and the preparation/cooking method), and to record the time and place where food was consumed. Two non-consecutive weekdays and a weekend day was recorded by participants. All dietary records were verified by the same registered dietitian either by phone or in person. The verification included checking all items in the dietary records and adding further details on foods and drinks (brands or recipes of dishes) consumed if necessary for the nutrient calculation. The dietary records were entered into the NutriComp food (nutrient) analysis program by registered dietitians [29]. Intakes of energy (kcal/day), salt (g/day), salt density (g/kcal), potassium (g/day), and potassium density (mg/kcal) were reported.

### 2.5. Statistical Analysis

Sample size was determined on the basis of the protocol, considering that samples of 200 people (100 per sex) were sufficient to characterize the group mean with 95% confidence intervals of ±12 mmoL/day (±276 mg/day), assuming a standard deviation of urinary sodium excretion of approximately 60 mmoL/day [34]. All statistical analyses were carried out with Stata 16.0 (StataCorp, College Station, TX, USA). For continuous variables, after testing for normal distribution (Kolmogorov-Smirnov and Shapiro-Wilk tests), either a Student’s *t*-test or the Mann-Whitney test (for non-normally distributed results) was used to assess differences between group means, and the Pearson chi-square test was performed to determine associations between categorical variables. The results are reported as mean (SD), median (IQ range) or percentages, as appropriate. A Pearson correlation coefficient was calculated to obtain information about the relationships between variables. We considered two-sided *p* < 0.05 as statistically significant.

## 3. Results

### 3.1. Characteristics of the Participants

The characteristics of the 189 participants in the final study are shown in Table 1. The mean age was 67.1 (SD 5.4) years. Considering the WHO body mass index (BMI) cut-points of 25 and 30 kg/m^2^ to classify overweight and obesity, respectively, 84% of men and 76% of women participants were classified as overweight or obese. Given the waist circumference cut-points of 102 cm for men and 88 cm for women, men and women participants with abdominal obesity constituted 58% and 69% of the sample, respectively. The prevalence of hypertension was 77% and 74% for the men and women, respectively. No statistically significant difference in the mean age; proportions of different age groups; systolic and diastolic BP; or the prevalence of overweight, obesity, abdominal obesity, and hypertension was found between the men and the women. The men had significantly higher BMI and waist circumference than the women.

### 3.2. Biochemical Characteristics

Considering the biochemical characteristics of the study participants, 36% had serum fasting glucose levels above the desirable level of 6 mmoL/L [35]. Mean serum cholesterol was 4.9 mmoL/L (SD 1.3), and 24% of men and 58% of women were above the 5.2 mmoL/L threshold. The serum mean LDL level was 3.1 mmoL/L (SD 1.2), the HDL level was 1.3 mmoL/L (SD 0.4) and the triglyceride level was 1.7 (SD 0.8) mmoL/L (Table 2).

### 3.3. Nutrient Intake Based on Three-Day Dietary Records

Table 3 shows the nutrient intake calculated from the three-day dietary records. The average daily energy intake was 2432 and 1910 kcal for men and women, respectively. Men consumed 13.4 g of salt on average (SD 4.0, median 12.6), which was significantly more than the women (9.9 g, SD 3.0, median 9.6). Salt intake was higher in men even after adjusting for energy intake. The WHO and nationally recommended levels of 5 g or less of salt was met by only 1 out of 189 female participants. No difference was found in potassium consumption between men and women, although the per calorie basis of women’s potassium consumption was higher than that of men. This result reflects that women have a more favorable sodium/potassium ratio than men (women: 2.43 vs. men: 3.07). Furthermore, 22% of elderly male and 16% of elderly female participants met the levels of potassium intake of 3.5 g/day recommended by national guidelines and the WHO, with no difference between sexes.

### 3.4. Sodium and Potassium Excretions

Table 4 describes sodium and potassium excretions, both adjusted and unadjusted to caloric intake. The average urinary volume excretion was 1888 mL/day, with no difference between sexes. Urinary creatinine excretion was 10.3 mmoL per day and was higher in men than in women. Mean urinary sodium was 213.4 (SD 79.3, median 210) mmoL/24 h for men and 161.7 (SD 55.5, median 155) mmoL/24 h for women. Assuming that 100% of dietary sodium is excreted in the urine, these values are equivalent to a mean consumption of 12.5 (SD 4.6, median 12.3) g of salt per day for men and 9.5 (SD 3.3, median 9.1) g of salt per day for women. Salt excretion was not different between men and women after adjusting for energy intake. The WHO and nationally recommended level of 5 g or less of salt was met by 7% of both male and female participants. Mean urinary potassium was 69.2 (SD 26.4, median 69) mmoL/24 h for men and 62.0 (SD 21.2, median 60) mmoL/24 h for women. Assuming that 85% of dietary potassium is excreted in the urine [18], these values are equivalent to a mean consumption of 3.18 (SD 1.21, median 3.18) g of potassium per day for men and 2.85 (SD 0.98, median 2.76) g of potassium per day for women. No statistical difference was found in potassium excretion between men and women. However, the per calorie basis of women’s excretion was significantly more than that in men. Furthermore, 34% of elderly male and 28% of elderly female patients met the potassium excretion level of 90 mmoL/day or more (equivalent to 3.5 g potassium) recommended by the national and WHO guidelines, with no statistical difference between sexes. The urinary molar sodium/potassium ratio was more favorable in women (2.81) than in men (3.25).

The estimation of salt intake based on three-day dietary records correlated poorly with the 24 h urinary sodium excretion measurement (r^2^ = 0.11, *p* = 0.15). Mean measured salt intake was on average 0.67 ± 4.5 g/day lower than estimated with the 3-day dietary records.

## 4. Discussion

This study is the first where 24 h urinary sodium and potassium excretion was used to estimate sodium and potassium intake in a large sample of Hungarian elderly. A high prevalence of CVD risk factors (hypertension: 76%, obesity: 36%, abdominal obesity: 63% of total; high cholesterol level: 25% of men and 58% of women) co-existed with excessive sodium and inadequate potassium intakes. The mean daily salt consumption calculated from 24 h urinary excretion was 11.0 g/day, which exceeded the recommended maximum intake of 5 g/day [10,19] in 93% of both men and women. The mean salt intake calculated from the 3-day dietary records was high, and exceeded the intake estimated from 24 h urine collection by 0.67 g. The mean potassium intake calculated from 24-h urine was 3.02 g/day (assuming 85% excretion into urine), which failed to meet the recommended level of 3.5 g/day [18,19] in 66% of men and 72% of women. The mean potassium intake calculated from the 3-day dietary records was 2.9 g/day, similar to the level estimated from 24 h urine. The sodium/potassium ratio is an important factor in the development of hypertension. The WHO recommendation for the sodium and potassium intakes are 2 g and 3.5 g, respectively, which translates into a Na/K molar ratio of approximately 1. In the current study, the Na/K molar ratios were 2.77 based on the 3-day dietary records; and 3.04 based on 24 h urine sodium and potassium excretion, both higher than the ratio calculated from the WHO sodium and potassium recommendations.

When interpreting results, salt intake estimates can be compared when obtained with the same method. In the current study, the estimated salt intakes based on three-day dietary records and 24 h urinary sodium excretion correlated poorly. Previous studies using dietary assessment methods usually report lower salt intake because they do not capture discretional salt consumption properly [26]. By contrast, in the current study, the salt intake based on 24 h urinary assessment was lower than that estimated from three-day dietary records. We assume that the food composition database updates of our nutrient analysis program could not keep up with the salt reduction that occurred recently in popular food items due to government-initiated public health interventions. Furthermore, the 3-day dietary records were carefully validated, and each potential salt source was clarified. If forgotten, the added salt intake was corrected via oral interview with a trained dietitian.

The results of the present study can be compared with available data on dietary salt intake in Hungary from national dietary surveys. The last two national representative surveys conducted with a similar methodology to the one used in this study report salt and potassium intake data based on 3-day dietary records. The mean salt intake of elderly men (age ≥ 65 years) was higher (14.3 and 14.4 g/day) and the potassium intake was lower (2.9 and 2.8 g/day) in 2009 and 2014, respectively, compared with the daily 13.4 g of salt and 3.0 g of potassium intake recorded in the present study for elderly (age > 60 years) men. For elderly women (age ≥ 65 years), salt intake was also higher (11.7 and 11.3 g/day), and potassium intake was lower (2.6 and 2.5 g/day) in 2009 and 2014, respectively, compared with the daily 9.9 g of salt and 2.8 g of potassium intake recorded in the present study for elderly (age > 60 years) women. Consequently, the sodium/potassium ratio in the present study was more favorable than that obtained in national diet surveys [24,25]. In this comparison, we considered that, unlike the samples in the national diet surveys, our sample was not representative of the Hungarian elderly population; nevertheless, the methods and analysis tools we used were very similar (3-day dietary record, same method and software for data cleaning and processing).

The Intersalt study conducted in 1985–86 was the first large-scale international study to publish comparable urine-based data on sodium and potassium consumption globally, with data from 52 centers worldwide [23]. The samples consisted of 200 men and women aged 20–59. In the Intersalt study, the mean urinary sodium excretion of the Hungarian sample was 232.2 (88.2) mmoL/24 h and 164.4 (58.5) mmoL/24 h; the mean urinary potassium was 59.0 (19.8) mmoL/24 h and 40.7 (11.6) mmoL/24 h; and the mean urinary creatinine was 13.3 (3.7) mmoL/24 h and 9.0 (2.0) mmoL/24 h; in men and women, respectively [36]. In the present study, the estimated sodium excretion in men was less than that measured in the Intersalt study by nearly 10%, but not different in women. Considering that the urine collection method used in the Intersalt study was very similar to that of the present study, one explanation of the decrease in sodium excretion in men can be the age difference between the two studies [37]. In the Biomarker2019 study, the participants were elderly (age > 60 years), whereas the Intersalt study recruited non-elderly adult participants (age 20–59 years). Data from the Hungarian Diet and Nutrition Status Survey-OTAP2014 show that elderly men (age ≥ 65 years), consume significantly less energy and as a result less sodium than the adult population (age 18–64 years), since sodium is evenly distributed in staple foods in Hungary (bread, meat products) [24,29]. On the other hand, the energy intake difference between elderly and adult women is less pronounced, only 7%, when the younger and middle-aged women groups are combined (age 18–64 years) in comparison with the elderly (age ≥ 65 years) in the same survey [24,29]. Therefore, the age difference between the Intersalt and the Biomarker2019 study presumably does not result in a significant difference in salt excretion for the population of women. Another source of difference in sodium excretion between the two studies can be changing dietary habits. According to the data from representative national dietary surveys, average daily salt consumption in the Hungarian adult population decreased from 19.2 g to 15.9 g in men and from 15.3 g to 11.2 g in women, between 1985 and 2014 [24,38]. Based on these data, a more significant sodium excretion reduction could be expected between 1985–86 (Intersalt study) and 2019 (Biomarker study). Potassium excretion in the present study was nearly 20% higher in men and more than 50% higher in women than that found in the Intersalt study. The age difference between the Intersalt and Biomarker2019 studies does not explain this increase, since the potassium intake is lower in the elderly men (age ≥ 65 years) and similar in elderly women compared to the corresponding adult age group (age 18–64 years) according to the Hungarian Diet and Nutrition Status Survey OTAP2014 [24]. The main food groups contributing to potassium intake are fruits and vegetables, meat products, and dairy [24]. The increase in potassium intake can be accounted for by the wider availability of fruits and vegetables in 2019 than in 1985. Urinary creatinine was lower in the Biomarker2019 study by nearly 10% in both men and women compared with the Intersalt study. As subjects with severe renal insufficiency were excluded from the present study, this change is probably attributable to the age difference between the two studies, as elderly people tend to have lower muscle mass.

Studies utilizing urinary sodium excretion for the assessment of salt intake in the elderly are limited, and only few used full 24 h urine collection. In these studies, age categories were not standardized. In the population >50 years old, the respective urinary sodium excretion levels in men and women are 201 and 125 mmoL/24 h in Ireland (mean for 65–74 years) [39], 183 and 152 mmoL/24 h in Austria (mean for 65–80 years) [40], 180 and 140 mmoL/24 h in Italy (mean for 65–74 years) [41,42], 180 and 125 mmoL/24 h in Switzerland (mean for ≥60 years) [43], 177 and 146 mmoL/24 h in Germany (median for 60–69 years) [44], 174 and 113 mmoL/24 h in the Netherlands (median for 50–70 years) [45], and 141 and 112 mmoL/24 h in the UK (mean for 50–64 years) [46]. Notably, the 213 and 162 mmoL/24 h mean urinary sodium excretion of Hungarian men and women in the present study are the highest among the corresponding values in Europe.

The WHO recommends urinary assessment to measure population sodium and iodine intakes, and dietary assessment appears to be accurate for estimating potassium intake [47]. The mean potassium intake based on dietary assessment ranges from 3.14 g in Italian men to 3.82 g in Swedish men and 2.56 g in French women to 3.26 g in Swedish women [48]. Hungarian elderly men in the present study have lower potassium (2.98 g) consumption than any of those reported in other European countries by the European Food Safety Authority, whereas Hungarian women are in the middle of the range with their 2.82 g intake [48].

Epidemiological studies suggest that the urinary sodium-to-potassium ratio may be a superior metric as compared to separate sodium and potassium values for determining the relation to blood pressure and cardiovascular disease risks [49]. In the Intersalt study, the mean 24 h urine Na/K molar ratio ranged from 0.01 (Yanomamo, Brazil) to 7.58 (Tianjin, China), and a 4.14 mean Na/K molar ratio was reported for the Hungarian sample [36]. In the Biomarker2019 study, the mean Na/K molar ratio was 3.04, more favorable than that reported for the Hungarian sample in the Intersalt study. There is no accepted recommended value for the Na/K molar ratio, but based on the World Health Organization published guidelines for sodium and potassium intake, the molar ratio is approximately 1. Iwahori and co-workers used data from the Intersalt study to analyze the associations of 24 h urinary Na/K ratios with 24 h urinary Na and K excretion. The Pearson’s correlation coefficients for the Na/K molar ratio of 24 h urine with Na and K excretions of 24 h urine were 0.57 and −0.48, respectively, in individuals [50]. In the present study, the relationships between the Na/K molar ratio and the Na and K excretions were weaker, the Pearson’s correlation coefficients for Na and K excretions were 0.5 and −0.4, respectively. The Na/K molar ratio in this study clearly presents an increased risk for high blood pressure and CVD for the elderly participants.

### 4.1. Strengths and Limitations

The important strength of this study is the use of the gold standard 24 h urine collection for dietary salt intake estimation. This method is unaffected by the recognized limitations of dietary recalls and records. According to the WHO protocol [27], we excluded participants with severe renal insufficiency and diuretic therapy which started within 2 weeks of the beginning of the study, further improving the accuracy of the salt intake estimation. The fieldworkers were registered nurses who underwent a one-day training. We provided oral directions, a written protocol, infographics, and collection containers to the study participants for urine collection. The dates and times of the collections were recorded. During the second visit, the registered nurses checked the collection for compliance with the protocol. If an incorrect method or spillage occurred, participants were either approached for re-collection or excluded from the study. Given this support, the compliance of the study was exceptionally good, attrition was minimal, and only 5% of the samples were excluded because of deviation from the urine collection protocol. Thus, for men, we almost reached (*n* = 99 subjects) the WHO recommended sample size of 100. Another strength is that we measured sodium and potassium intake concurrently within the same study, using both diet- and urine-based assessment methods. Further improving the quality of the study was that all of the clinical laboratory measurements were carried out in the same laboratory.

A limitation is that the study results cannot be generalized to the entire elderly population. Although the study included a heterogeneous population of elderly men and women from different age categories with quota sampling, this population does not represent the entire Hungarian population as we managed to recruit participants only from six settlements in western Hungary. Nevertheless, we plan to use the established 24 h urine collection protocol in a subsample of the forthcoming representative National Diet and Nutrition Status Surveys. In addition, our sample size for women was 10% smaller than that recommended by the WHO protocol [27]. Another limitation of our study is the use of a single 24 h urine collection per individual. This approach is valid and reliable for estimating the population levels of sodium and potassium intake, but is poor at an individual level because of marked day-to-day variation [51].

### 4.2. Policy Implications

Hungary joined the European Salt Reduction Framework Program in 2008. Within the framework of the program, Hungary pledged to reduce the salt content of processed food by an average of 16% over 4 years and to implement the key elements of the program. The STOP SALT National Salt Reduction Program was announced in 2010 and consists of three elements, namely, raising campaigns, negotiating with the industry to reduce salt intake in foodstuffs, and monitoring the salt intake of the population. Our public information campaign took place in 2010. The analysis of the National Diet and Nutrition Status Survey in 2009 revealed that among the processed foods, bread, meat products, and preserved vegetables and pickles account for 36%, 21% and 10% of salt intake, respectively [25]. The high salt intake from bread prompted a negotiation with the Hungarian Bakers’ Association, and a pledge to gradually reduce the salt content of bread by 16% from 2012 to 2017 was signed. In the framework of the program, salt intake monitoring was conducted with the National Diet and Nutrition Status Survey 2014 by using 3-day dietary records. From the beginning, the Hungarian Hypertension Society helped raise awareness of the program as a key partner, and other member companies such as the National Heart Foundation, are also cooperating actively. In 2014, the food industry started a salt reduction initiative, and several companies joined with pledges to reduce the salt contents of their manufactured food. Another important intervention affecting the salt content of food was the Public Heath Product Tax, which was levied to food which are high in salt (salty snacks and condiments). In 2013, standards were introduced in public catering, which set a limit to the maximum amount of salt per day in meals provided by public catering. This standard could affect the salt intake of a wide population because in small settlements, local school kitchens often provide food for the elderly. The effects of these interventions on a population’s salt intake are difficult to evaluate because the monitoring is based on 3-day dietary records, which have considerable limitations. Nevertheless, when the salt consumption results of the dietary survey in 2009 were compared to those in 2014, no significant reduction in salt intake was detected [24,25]. Considering the highly unfavorable health statistics and that the population’s salt intake is almost three times the WHO recommended value, the need to continue the salt reduction program to reduce high blood pressure, cardiovascular disease and stroke is unquestionable. Furthermore, a reliable monitoring system is required to assess the influence and soundness of the program.

## 5. Conclusions

This study provided the first objective measure of sodium and potassium intake in a Hungarian elderly sample by using the gold standard 24 h urinary excretion method. Compared with recommendations, the dietary salt intake assessed by urinary sodium excretion (gold standard) and by 3-day dietary records was higher, and potassium intake was lower. The coexistence of these factors poses a cardiovascular risk to this Hungarian elderly population. The findings underline the need to intensify salt reduction efforts in Hungary.

## Figures and Tables

**Table 1 ijerph-18-08806-t001:** Demographic and anthropometric characteristics of study participants, Biomarker2019 survey.

Variable	All *n* = 189	Men *n* = 99	Woman *n* = 90	*p* *
Age (years) ^#^	67.1 (5.4)	66.6 (5.5)	67.7 (5.2)	0.06
Age groups ^##^				
60–64 years N (%)	68 (36%)	41 (41%)	27 (30%)	0.20
65–69 years N (%)	73 (39%)	37 (37%)	36 (40%)	
70+ years N (%)	48 (25%)	21 (21%)	27 (30%)	
BMI (kg/m^2^) ^#^	28.9 (4.6)	29.7 (4.5)	28.1 (4.5)	0.02
Waist circumference (cm) ^#^	99.6 (13.1)	105.0 (12.3)	93.7 (11.3)	<0.001
Systolic blood pressure (mm Hg) ^###^	132.1 (14.2)	133.1 (13.6)	131.0 (14.7)	0.31
Diastolic blood pressure (mm Hg) ^###^	76.1 (8.2)	76.8 (7.2)	75.2 (9.1)	0.18
Overweight N (%) ^## §^	84 (44%)	43 (43%)	41 (46%)	0.15
Obese N (%) ^## §§^	68 (36%)	41 (41%)	27 (30%)	
Abdominal obesity N (%) ^## +^	119 (63%)	57 (58%)	62 (69%)	0.11
Hypertension N (%) ^# %^	143 (76%)	76 (77%)	67 (74%)	0.71

Results are mean (SD) or N (%), ^#^ Mann-Whitney test, ^##^ Pearson chi^2^ test, ^###^ Student’s *t*-test, * comparison between sexes, ^§^ ≥ 25 BMI < 30; ^§§^ BMI ≥ 30, ^+^ waist circumference ≥102 cm in men and ≥88 cm in women, ^%^ hypertension diagnosed by medical doctor (questionnaire).

**Table 2 ijerph-18-08806-t002:** Biochemical characteristics of study participants, Biomarker2019 survey.

Variable	All *n* = 189	Men *n* = 99	Woman *n* = 90	*p* *
Serum glucose (mmoL/L) ^#^	6.1 (1.8)	6.5 (2.1)	5.6 (1.2)	0.0012
Serum cholesterol (mmoL/L) ^##^	4.9 (1.3)	4.5 (1.2)	5.4 (1.3)	<0.001
LDL cholesterol (mmoL/L) ^##^	3.1 (1.2)	2.7 (1.0)	3.5 (1.2)	<0.001
HDL cholesterol (mmoL/L) ^#^	1.3 (0.4)	1.2 (0.3)	1.5 (0.4)	<0.001
Serum triglyceride (mmoL/L) ^#^	1.7 (0.8)	1.9 (1.0)	1.6 (0.7)	0.1138
Serum uric acid (µmol/l) ^#^	332.3 (76.6)	354.4 (73.3)	308.0 (73.0)	<0.001

Parameters were measured from fasting blood. Results are mean (SD), ^#^ Mann-Whitney test, ^##^ Student’s *t*-test, * comparison between sexes.

**Table 3 ijerph-18-08806-t003:** Energy, salt and potassium intakes from three-day dietary records and the proportion of participants meeting recommended targets for salt and potassium intake, Biomarker2019 survey.

Variable	All *n* = 189	Men *n* = 99	Woman *n* = 90	*p* *
Energy (kcal/day) ^#^	2183 (605) 2088 (1770–2553)	2432 (625.9) 2387 (2006–2775)	1910 (443.1) 1843 (1587–2138)	<0.001
Salt (g/day) ^#^	11.7 (4.0) 11.0 (8.9–13.5)	13.4 (4.0) 12.6 (10.6–16.1)	9.9 (3.0) 9.6 (7.8–11.5)	<0.001
Sodium (mmoL/day)	200.2 (67.7) 187.5 (152.3–231.0)	228.5 (68.9) 215.3 (181.9–276.0)	169.0 (50.7) 163.7 (134.1–196.5)	<0.001
Salt (energy adjusted) (g/1000 kcal/day) ^#^	5.4 (1.3) 5.4 (4.6–6.2)	5.5 (1.1) 5.7 (4.7–6.3)	5.3 (1.5) 5.2 (4.3–6.0)	0.0083
Salt intake <5g/day N (%) ^###^	1 (0.5%)	0 (0%)	1 (1.1%)	0,476
Potassium (g/day) ^#^	2.90 (0.72) 2.85 (2.36–3.35)	2.98 (0.75) 2.96 (2.39–3.46)	2.82 (0.68) 2.74 (2.33–3.27)	0.1551
Potassium (mmoL/day)	74.5 (18.6) 73.1 (60.6–85.9)	76.4 (19.3) 75.6 (61.3–88.7)	72.3 (17.5) 70.2 (59.8–83.8)	0.1339
Potassium (energy adjusted) (g/1000 kcal/day) ^#^	1.4 (0.4) 1.3 (1.2–1.6)	1.3 (0.3) 1.2 (1.1–1.4)	1.5 (0.4) 1.5 (1.3–1.8)	<0.001
Potassium intake >3.5g/day N (%) ^####^	36 (19.0%)	22 (22.2%)	14 (15.6%)	0.244
Na/K molar ratio ^##^	2.77 (0.87) 2.75 (2.16–3.36)	3.07 (0.83) 3.03 (2.37–3.71)	2.43 (0.80) 2.30 (1.91–2.93)	<0.001

Results are mean (SD) and median (25th–75th percentile) or N (%), ^#^ Mann-Whitney test, ^##^ Student’s *t*-test, ^###^ Fisher’s exact test, ^####^ Pearson chi^2^ test, * comparison between sexes.

**Table 4 ijerph-18-08806-t004:** Daily urinary excretions of volume, sodium, potassium and creatinine; estimates of salt and potassium intake; and proportion of participants meeting recommended targets for salt and potassium consumption, Biomarker2019 survey.

Variable	All *n* = 189	Men *n* = 99	Woman *n* = 90	*p* *
Volume (ml/24h) ^#^	1888 (578) 1831 (1520–2242)	1893 (584) 1924 (1456–2242)	1882 (574) 1765 (1551–2259)	0.7093
Creatinine (mmoL/24h) ^#^	10.3 (3.8) 9.3 (7.5–12.3)	12.4 (4.0) 11.8 (9.3–14.6)	8.0 (1.8) 8.0 (6.7–8.9)	<0.001
Sodium (mmoL/24h) ^#^	188.8 (73.5) 176 (137–233)	213.4 (79.3) 210 (156–269)	161.7 (55.5) 155 (130–191)	<0.001
Sodium (energy adjusted ^+^) (mmoL/1000 kcal/24h) ^##^	89.8 (36.9) 85.5 (67.3–105.3)	91.4 (37.1) 92.9 (69.6–106.8)	88.0 (36.9) 79.8 (65.5–103.7)	0.1339
Salt intake (g/day) ^# ß^	11.0 (4.3) 10.3 (8.0–13.6)	12.5 (4.6) 12.3 (9.1–15.7)	9.5 (3.3) 9.1 (7.6–11.2)	<0.001
Salt < 5g/day N (%) ^#### ß^	13 (6.9)	7 (7.1)	6 (6.7)	0.9130
Potassium (mmoL/24h) ^#^	65.8 (24.3) 65 (49–81)	69.2 (26.4) 69 (51–83)	62.0 (21.2) 60 (46–79)	0.0618
Potassium (energy adjusted ^+^) (mmoL/1000 kcal/day) ^#^	32.4 (16.5) 28.8 (21.6–38.3)	30.6 (17.7) 27.5 (20.2–35.8)	34.3 (15.0) 31.7 (24.3–41.3)	0.0386
Potassium intake (g/day) ^# ßß^	3.02 (1.11) 2.98 (2.25–3.72)	3.18 (1.21) 3.17 (2.34–3.81)	2.85 (0.97) 2.75 (2.11–3.62)	0.0618
Potassium intake >90 mmoL/day N (%) ^####^	59 (31.2)	34 (34.3)	25 (27.8)	0.331
Na/K molar ratio	3.04 (1.17) 2.88 (2.25–3.63)	3.25 (1.23) 3.17 (2.36–3.92)	2.81 (1.06) 2.67 (2.22–3.41)	0.0084

Results are mean (SD) and median (25th–75th percentile) or N (%), ^#^ Mann-Whitney test, ^##^ Student’s *t*-test, ^####^ Pearson chi^2^ test, * comparison between sexes, ^+^ energy intake calculated by three-day dietary records, ^ß^ grams of salt = mmoL Na × 23 × 2.5421/1000, assuming 100% sodium excreted in 24 h urine collection, ^ßß^ grams of potassium = (mmoL K × 39/1000) × 1.18, assuming 85% dietary potassium excreted in the urine.

## Data Availability

Data available on request due to privacy restrictions. The data presented in this study are available on request from the corresponding author.

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
