# Peer review of "Dietary Sodium and Potassium Intake in Hungarian Elderly: Results from the Cross-Sectional Biomarker2019 Survey"

_ijerph, 2021, doi:10.3390/ijerph18168806_

Round 1

Reviewer 1 Report

In the submitted manuscript, the authors have presented an approach aimed at estimating sodium and potassium consumption in Hungarian elderly subjects. Sodium/potassium intake was estimated by using the traditional three-day dietary records and the newly established 24-hour urine collection, which is considered the gold standard for salt intake estimation.

This is an interesting article. The reported methods of analysis appear appropriate, and the results do not appear to be over-interpreted. The authors have correctly highlighted the strengths and limitations of the study.

I found the paper to be overall well written and much of it to be well described.

Reviewer 2 Report

  1. The statement in lines 66-67, “A Hungarian survey of sodium and potassium intake via 24-hour urine collection has been lacking thus far” is incorrect. Actually, various biomarkers were obtained in this study and this enables to compare with the study findings from INTERSALT study (see PMID: 2810329 and supplemental Table of PMID: 30996260). INTERSALT study was conducted in 1985-86 and reported that the mean urinary sodium was 232.2 (88.2) mmol/24h and 164.4 (58.5) mmol/24h in men and women; the mean urinary potassium was 59.0 (19.8) mmol/24h and 40.7 (11.6) mmol/24h in men and women; the mean urinary creatinine was 13.3 (3.7) mmol/24h and 9.0 (2.0) mmol/24h in men and women; and the mean urine volume was 1.73 (0.65) l/24h and 1.14 (0.44) mmol/day in men and women in Hungarian population (ages 20-59). It seems that sodium excretion decreased by nearly 10% in men but remained similar; potassium excretion increased by nearly 20% in men and more than 50% in women; urinary creatinine decreased by nearly 10% in both men and women; and urine volume increased nearly 10% and 70% in men and women in past 35 years. Please speculate what made these differences from lifestyle change in Hungarian population (e.g., dietary habit) and from methodological (e.g., urine collection procedures) perspectives.

  1. In lines 53-60, the authors pointed out that “Low dietary potassium can increase the effect of sodium on BP, and the relationship between sodium and BP is strengthened if urinary sodium/potassium ratio is considered” and “In light of the cardiovascular implications of consuming too much sodium and too little potassium, an important public health initiative is to monitor the intake of these nutrients.” Findings from the INTERMAP study demonstrated that the relation of dietary sodium to BP with potassium has been clarified in PMID: 29507099. Furthermore, Na/K Ratio is a surrogate index for higher Na intake and lower K intake as introduced in PMID: 28678188; findings from INTERSALT study demonstrated that the change in the urinary Na/K molar ratio from 3.09 to 1.00 delivers 3.36 mmHg of estimated reduction in the population systolic BP and the estimated reduction was larger for the Na/K ratio compared to when the Na and K were analyzed separately. Na/K ratio shows better estimate for 24-hour urine from multiple spot urine specimens compared to Na and K separately (see PMID: 24718298, 26310187, 28039381, PMID: 30443006). Also, self-monitoring device may become available in near future (see PMID: 29093302). Why not prioritize Na/K ratio in this manuscript and deprioritize Na and K? Separate Na and K are weaker index in the relation to BP and cardiovascular outcomes, and less accurate for estimating 24-hour urine form spot urine which derives a paradoxical relationship regarding the Na intake vs. the BP and CVD (see PMID: 27248297).

  1. The result shown for Na/K ratio in this study seem to be presented not in molar ratio but in gram ratio. Most of the value for Na/K ratio in medical journal are shown in Na/K molar ratio, but Na/K ratio in gram in nutrition journals. To avoid confusion, I would suggest Na/K molar ratio for this journal. Same applies for sodium and potassium intake.

Round 2

Reviewer 2 Report

1. In lines 343-345, the authors referred to the dietary survey methodology. However, I don't see improvement and changes of the dietary survey methodology between these 2 studies since 24-hour urine collection have been the gold standard throughout several decades and reliable new suggested alternative method has not become available yet. Moreover, the older INTERSALT study protocol may have been rather strict than this study. Please clarify what this statement means since uneasy to understand what this stands for.

2. In lines 392-394, the reviewers made a statement that the lowest CVD risk was found between Na/K molar ratios 1 and 2. However, this maybe by reverse causality. Ill patients may restrict Na intake hard to survive (That's why the risk is not lowest at the lowest Na/K ratio). Thus, it is inappropriate to conclude that J-curve is the truth of the association with CVD risk. Please modify the related statements.

3. As the authors replied, I understood the aim of this study adding data on national sodium and potassium intake estimates on population level in Hungary. PMID: 30996260 demonstrates the association between urinary sodium-to-potassium ratio and intake of sodium and potassium. Please speculate association between urinary sodium-to-potassium ratio and Na intake by the data obtained in this study and referencing PMID: 30996260.
